# Phages in Food Industry Biocontrol and Bioremediation

**DOI:** 10.3390/antibiotics10070786

**Published:** 2021-06-28

**Authors:** Pablo Cristobal-Cueto, Alberto García-Quintanilla, Jaime Esteban, Meritxell García-Quintanilla

**Affiliations:** 1Department of Clinical Microbiology, IIS-Fundación Jiménez Díaz, Av. Reyes Católicos, 2, 28040 Madrid, Spain; pablourjc09@gmail.com (P.C.-C.); jesteban@fjd.es (J.E.); 2Department of Biochemistry and Molecular Biology, School of Pharmacy, University of Seville, Calle Profesor García Gonzalez, 2, 41012 Seville, Spain; albertgq1970@us.es

**Keywords:** bacteriophage, food industry, bioremediation, biocontrol, animal, plant, surface

## Abstract

Bacteriophages are ubiquitous in nature and their use is a current promising alternative in biological control. Multidrug resistant (MDR) bacterial strains are present in the livestock industry and phages are attractive candidates to eliminate them and their biofilms. This alternative therapy also reduces the non-desirable effects produced by chemicals on food. The World Health Organization (WHO) estimates that around 420,000 people die due to a foodborne illness annually, suggesting that an improvement in food biocontrol is desirable. This review summarizes relevant studies of phage use in biocontrol focusing on treatments in live animals, plants, surfaces, foods, wastewaters and bioremediation.

## 1. Introduction

The World Health Organization (WHO) estimates that around 420,000 people die every year due to a foodborne illness. This has an economic impact of US$110 billionon the global economy. Furthermore, according to the WHO, approximately 18% of the infectious disease outbreaks are related to the water in Europe [1], and this percentage may be higher in other continents. On the other hand, multidrug resistant (MDR) bacteria are a big concern not only in human health, but also in livestock industries. The global estimation of antimicrobial consumption (mg) per population correction unit (PCU) reported for cattle, chickens and pig is 45 mg/PCU, 148 mg/PCU and 172 mg/PCU, respectively, and a rise of antimicrobials in food animal production has been projected by 67% from 2010 to 2030, reaching 105,596 (±3605) tons of antimicrobials by 2030 [2]. In this sense, China consumes the largest quantity of antimicrobials, followed by the United States of America, Brazil, Germany and India [2]. Moreover, transmission of MDR bacteria from animals to humans has been described [3] and is subject of surveillance.

In the last decades, the use of bacteriophages has reappeared in Western countries as an alternative to chemicals treatments [4]. Bacteriophages are recognized as the most abundant biological agents on Earth, due to their ubiquitous presence in the environment. Phages, for short, are able to lyse MDR bacteria and reduce the non-desirable effects produced by chemicals on food. According to their life cycle, they can be classified as virulent phages (lytic phages) or temperate phages (lysogenic phages) [5]. Lytic phages use the genomic and biosynthetic machinery of the bacteria to produce their progeny, provoking the bacterial lysis and their consequent release. The phage-encoded endolysins are ultimately responsible to break down the bacterial peptidoglycan at the final stage of the cycle [6]. Conversely, lysogenic phages are capable of incorporating their nucleic acid into the genome of the host cell or just remain like a plasmid into the host cell during multiple bacterial generations. Therefore, only lytic phages are usually used in bioremediation or phage therapy. Bacteria can become resistant to phages by modifying their receptors, turning them inaccessible or non-complementary to the phage receptor binding protein [7]. Fortunately, these insensitive strains can be lysed using cocktails of phages instead of a single phage.

Phages in food industry can be applied at different stages [8]: directly on animals or plants to eliminate the probability of bacterial infection and disease, in food production plants to prevent bacterial biofilm formation, or directly on food to preserve the product. This review excludes in vitro experiments and summarizes selected in vivo findings of phage use in non-human biocontrol, focusing on the treatment of live animals and plants that are relevant in the food industry, as well as the raw food products, and the biofilm control on surfaces during their processing and manufacturing, ending with the bioremediation of the wastewaters generated (Figure 1).

## 2. Food and Phages

According to the Centers for Disease, Control and Prevention (CDC) [9], foodborne infections produced by chicken, beef, pork and turkey are associated with *Campylobacter* and *Salmonella* presence meanwhile dairy products like raw milk and cheese are commonly infected by *Campylobacter*, *Salmonella*, *Escherichia* and *Listeria.* Most common bacteria infecting vegetables and fruits are *Salmonella*, *Escherichia* and *Listeria* and most frequent pathogen producing foodborne illnesses in fishes and shellfishes are *Vibrio* and *Salmonella*.

### 2.1. Bacteriophages to Control Salmonella enterica

One in three foodborne outbreaks in the European Union in 2018 were caused by *Salmonella*, being salmonellosis the second most commonly reported gastrointestinal infection in humans (91,857 cases reported) after campylobacteriosis (246,571) [10].

Thung et al. studied the interaction of the bacteriophage SE07, isolated from retail meat samples, against *S. enterica* serovar Enteritidis on different food matrices, such as fruit juice, fresh egg, beef and chicken meat. The reduction of the bacteria population in all of them was significant at 12 h (2.05 log CFU/mL, 1.98 log CFU/mL, 1.79 log CFU/mL, and 1.83 log CFU/mL, respectively), and after that time there was no further significant reduction [11]. In 2018, Phongtang et al. evaluated the effect of P22 phage (ATCC 97541) against *S. enterica* serovar Typhimurium in milk. This phage showed an inhibitory effect of more than 3 log UFC/mL reduction after 4 h [12]. Bao et al. tested two lytic phages, vB_SenM-PA13076 (PA13076) and vB_SenM-PC2184 (PC2184), in chicken breast, pasteurized milk and Chinese cabbage. Phages were isolated from chicken sewage and infected *S. enterica* serovar Enteritidis. PA13076 was able to infect 222 strains (71.4%) and PC2184 infected 298 strains (95.8%) out of 311 isolates tested. The two phages were rapidly inactivated at temperatures above 60 °C (PA13076) or 70 °C (PC2184). Interestingly, PA13076 reduced *Salmonella* population in chicken breast, pasteurized milk and Chinese cabbage by 2 log, 2 log and 2.5 log UFC/mL, respectively, whereas PC2184 reduced bacteria population in chicken breast, pasteurized milk and Chinese cabbage by 3 log, 4 log and 3.5 log UFC/mL, respectively [13].

The company Micreos Food Safety has developed the brand Phageguard S (Table 1) based on phages Felix-O1a and S16 against *Salmonella enterica*. This product is able to kill all *Salmonella* serovars including those that are resistant to antibiotics and the 20 most virulent *Salmonella* strains according to the United States Department of Agriculture (USDA). Phageguard S can reduce bacterial population by 1–3 log CFU/mL without affecting taste, odor or texture of foods. It is effective from 0 to 35 °C and its use is recommended as final treatment in spray or directly immersing food into the phage solution [14]. Yeh et al. reported that the combination of phages S16 and Felix-O1a reduced *Salmonella* on ground beef and pork by 1 and 0.8 log CFU/g, respectively [15]. A recent study tested Phageguard S on lean pork, bacon and pork trims with good results. The product administration decreased *Salmonella* population by 0.8–1.7 log CFU/cm^2^ or g using 5 × 10^7^ PFU/cm^2^ or g of phages [16].

### 2.2. Bacteriophages to Control Listeria monocytogenes

*L. monocytogenes* is peculiar due to its ability of growing at refrigerated temperatures (2–8 °C). Guenther et al. showed the effect of the lytic A511 phage to control *L. monocytogenes* in different ready to eat foods. In liquid samples as chocolate milk and mozzarella cheese brine, this phage was able to reduce the *L. monocytogenes* population below the detection limit, while in solid samples (hot dogs, sliced turkey meat, smoked salmon, seafood, sliced cabbage, and lettuce leaves) the reduction was above 5 log units [17]. Another phage used to control *L. monocytogenes* in food products and food processing environments is P70, a phage known to have a broad host range infecting *Listeria* sp. serovars 1/2a, 1/2b, 1/2c, 4a, 4c, 4d, 4e, 5, 6a and 6b with results over 62% of lysis [18].

Currently, there are two products based on phages approved in the US to be used in food industry against *Listeria* (Table 1). The United States Food and Drug Administration and the U.S. Department of Agriculture approved ListShield^TM^ (Intralytix, Baltimore, MA, USA) as a food additive for ready-to-eat meat and poultry products, usually as a spraying or dipping suspension [19]. ListShield^TM^ is a mixture of six lytic phages targeting *L. monocytogenes* that does not affect the organoleptic quality of foods and does not produce adverse effects on commensal microbiota [20]. Gutierrez et al. tested the product ListShield^TM^ on Spanish dry-cured ham and the surfaces that are commonly used in food industry and obtained a 100% lysis of *L. monocytogenes* strains examined. In dry-cured ham, the reduction of bacterial population was of 3.5 log units after 14 days of incubation at 4 °C. Moreover, ListShield was effective in removing 72 h biofilms formed on stainless steel surfaces by most of the assayed strains after four hours of treatment at 12 °C [21]. A recent study also tested the effectiveness of ListShield^TM^ in chicken breast. The phage treatment reduced the bacterial population 0.84 log CFU/mL when it was applied alone and 2.04 log CFU/mL in combination with UV-C treatment during storage for 72 h without significant differences in colour, pH or food quality [22].

The second formulation approved in the USA is LISTEX™P100 (Micreos Food Safety, Wageningen, The Netherlands), a brand composed of bacteriophage P100 produced to control *L. monocytogenes*. This product has been shown to reduce at least 3.5 log units on soft cheese [23]. Soni et al. demonstrated its activity on fresh channel catfish fillets (*L. monocytogenes* reduction between 1.4 and 2.0 log CFU/g at 4 °C, 10 °C, and 22 °C) [24], raw salmon (bacterial reduction of 1.8, 2.5, and 3.5 log CFU/g from initial bacterial loads of 2, 3, and 4.5 log CFU/g, respectively, at 4° and 22 °C) [25], and on soft cheese (with initial bacterial reduction of 2–4 log CFU/cm^2^ at 4 °C, but subsequent bacterial regrowth reported) [26]. Also this bacteriophage has been tested to reduce *L. monocytogenes* biofilms on stainless steel coupon surfaces resulting in high elimination of biofilm mass in all *L. monocytogenes* strains tested [27]. In 2017, the effect of this product was tested in sushi. Promising results were obtained in assays with initial 6-log CFU/g of bacteria and 8-log PFU/g of phage inoculation at 22 °C; a maximum reduction of 4.44 log CFU/g was achieved when the product was inoculated directly in sashimi samples, compared with the control group [28]. LISTEX™P100 has also been tested in soft cheeses achieving a reduction of more than 2 log CFU/mL [29]. Recently, the effect of the phage P100 in combination with the antimicrobial peptide pediocin PA-1 and mild high hydrostatic pressure was evaluated as a new method to eradicate *Listeria* from milk. The combination decreased immediately the *L. monocytogenes* population, although in a few cases a regrowth during the storage process was encountered [30].

### 2.3. Bacteriophages to Control Escherichia coli

The presence of *E. coli* in fruits, vegetables or animal products is a signal of inadequate hygiene during the processing methods in food industry since this bacterium is an indicator of fecal contamination in food and drinking water [31]. *E. coli* infections are characterized by diarrheal illnesses produced mainly by two strains: Shiga toxin-producing *E. coli* (STEC) and enterotoxigenic *E. coli* (ETEC) [32]. The detection of food contaminated by bacteria is actually a crucial strategy to avoid a large number of infections. For this reason, the use of bacteriophages is being implemented to detect these bacteria and their subsequent elimination.

In 2020, Duc et al. discovered the first phage able to reduce the population of three different bacteria: *E. coli* O157:H7, *S. enterica* serovar Enteritidis, and serovar Typhimurium. This phage decreased the population in chicken food by more than 1.3 log CFU/mL after a 2 h treatment at 4 °C and 24 °C [33]. Zampara et al. fused T5 endolysin and RBP Pb5 (which binds to the bacterial outer membrane ferrichrome transporter FhuA) in different configurations and showed that one of these innolysins named Ec21 was able to reduce *E. coli* by 2.2 log CFU per unit. Interestingly, innolysin Ec21 also displayed bactericidal activity against *E. coli* resistant to third-generation cephalosporins, reaching a 3.31 log reduction in cell counts [34].

### 2.4. Bacteriophages to Control Campylobacter sp.

*C. jejuni* and *C. coli* are frequent causes of human enteritis around the world. People can get infected with these bacteria by eating contaminated seafood, meat and undercooked poultry products.

Zampara et al. identified phages able to reduce *C. jejuni* at chilled temperature on contaminated poultry meat. These phages were dependent on capsular polysaccharides (CPSs) for infection, but they reduced bacterial population by at least 0.55 log CFU. The capacity of the two most bactericidal phages was better when combined in a cocktail, obtaining a reduction of 0.73 log CFU [35]. Recently, bacteriophage CJ01 has been tested as a biocontrol agent against *C. jejuni* in mutton and chicken meat. A reduction of 1.70 log CFU/g and 1.68 log CFU/g was obtained in treated mutton and chicken meat, respectively, at 4 °C [36].

### 2.5. Bacteriophages to Control Vibrio sp.

*Vibrio* sp. is found in tissues and/or organs of various marine algae and animals, like abalones, bivalves, corals, fish, shrimp, sponges, squid, and zooplankton. The CDC estimates that *Vibrio* causes approximately 52,000 foodborne illnesses and 100 deaths in the US every year [37].

A recent study reported that the VVP001 phage specifically infected *V. vulnificus* in a broad range of temperatures ranging from −20 °C to 65 °C, showing a reduction of 3.87 log CFU of bacteria on seafood [38]. Zang et al. showed that the OMN phage inactivated 90% and 99% of *V. parahaemolyticus* on oyster meat surface after 48 and 72 h, respectively, when it was applied directly on meat [39]. Jun et al. isolated and tested the pVp-1 phage against the pandemic multidrug resistant *V. parahaemolyticus* strain named CRS 09-17. Oysters were treated with a 72 h immersion with the phage and the bacterial reduction was of 4 log compared to the control group, while the direct treatment on the oyster surfaces reduced the CFU of bacteria by 6 log [40].

## 3. Phage Therapy for Animals

Phage administration is an interesting alternative to antibiotics in animals. Many in vitro experiments against pathogenic bacteria infecting animals have been reported. Here we focus on recent in vivo studies to show the state of the art in this field (Table 2).

In farms, phage therapy has been studied to treat and prevent infections caused by *Salmonella enterica* serovar Kentucky and *Escherichia coli* in chickens. Two *S. enterica* serovar Kentucky and three *E. coli* O119 phages were able to reduce mortality from 30% in positive control groups up to 0% in treated chickens. Notably, the higher reduction of bacteria counts in cecum, heart and liver was obtained at day 23 [41].

Recently, a bacteriophage cocktail was used against *Pseudomonas aeruginosa* that produces rhinosinusitis in sheep. A mix of four phages was able to reduce biofilm biomass on frontal sinus mucosa at concentrations of 10^8^–10^10^ PFU/mL with no safety concerns [42].

Several murine mastitis models have showed that phage therapy could be also used against *Staphylococcus aureus* in bovine mastitis caused by microbial infection [43,44,45]. A previous study published in 2006 by Gill et al. analyzed the efficacy of a 5-day treatment consisting of phage K administered intramammary in lactating Holstein cows with subclinical mastitis caused by *S. aureus*. Three out of 18 animals were cured (16.7%) compared to none out of 20 cows of the negative control group (0%) [46]. Despite some success, the low efficacy could be explained by the data of Gill et al. showing that incubation of *S. aureus* with whey and bovine serum resulted in inhibition of phage K lysis. Accordingly, they concluded that proteins could block sterically the phage K attachment to the bacteria, suggesting that *S. aureus* could be more resistant to phages in vivo in mastitis infections than in vitro experiments [47].

Infections caused by *E. coli* O157:H7 and treatment with phage therapy in ruminants have been already reviewed [48], revealing that further understanding of phage administration, effective multiplicity of infection (MOI) and correct analysis of results are necessary in cattle phage therapy [49,50,51]. In sheep, no significant reductions of *E. coli* O157:H7 were found compared to controls when a single phage was administered after oral *E. coli* inoculation [52,53]. However, a mix of two phages reduced more than 99% the presence of *E. coli* in the lower intestinal tracts of treated animals [54]. In addition, a cocktail of eight phages reduced significantly fecal *E. coli* O157:H7, although not in the rumen, after 24 h post phage administration [55].

In piglet studies, phages were able to kill methicillin-resistant *S. aureus* (MRSA) in vitro but no reduction was observed in the nasal mucosa in vivo or ex vivo [56]. This fact emphasizes the importance of considering other factors that may counteract phage efficacy in vivo, such as reduced adherence or increased clearance by the animal fluids. However, experiments conducted in growing pigs showed that dietary supplementation with a commercial cocktail of phages against *Salmonella*
*enterica*, *S. aureus*, *E. coli* and *Clostridium prefringens* was more efficient than probiotics as growth promoters [57], improving food digestibility, daily weight gain and gain per feed, among other parameters.

The presence of wounds is relatively common in swine. An hydrogel containing phages against *Acinetobacter baumanii* was used to reduce wound infections in an ex vivo model of pig skin, and achieved a 90% reduction in bacterial counts after only 4 h of treatment [58].

Another study showed that seven phages isolated from pig farms in the United Kingdom were able to lyse all 68 *Salmonella* strains tested, including MDR ones, offering a valuable alternative to antimicrobials to reduce infections and food poisoning [59].

Another recent review [60] summarized the known phages infecting *Paenibacillus larvae*. This spore-forming bacterium attacks honeybee larvae causing the American foulbrood, which is the most widespread and destructive of the honeybee brood diseases, being able to destroy an entire colony in just three weeks. Importantly, all known bacteriophages against *P. larvae* to date are lysogenic. Despite that, studies of phage therapy in vitro and in hives have shown higher survival rates of treated groups including prophylactic benefits. Lack of success in some cases was attributed to the lysogenic nature of the phages or their inability to access the gut.

In aquaculture, the common carp has been used as a model to demonstrate the effectiveness of phage therapy against *Citrobacter freundii*, using a single phage, IME-JL8. This bacterium belongs to the normal flora of fishes; however, it has been associated to systemic infection in common carp and other diseases in diverse fishes. Administration of phages into the carp decreased pro-inflammatory cytokines and protected the fish from infection when phages were administered one hour after bacteria inoculation, but not after 24 h, indicating that timing is relevant in phage therapy [61]. Similarly, no adverse inflammatory response was induced by the ETP-1 phage in zebrafish (*Danio rerio*), and twelve days of exposition to ETP-1 was able to increase survival from 18% in the control group up to 68% after infection with *Edwardsiella tarda* bacteria [62]. Another example can be found in the North African catfish (*Clarias gariepinus*). Ulcerative lesions caused by *P. aeruginosa* in North African catfish were reduced seven fold compared with untreated control after 8–10 days of treatment with a single phage [63]. In addition, treatment with two different phages at MOI of 100 reached 100% of survival in Vietnamese striped catfish (*Pangasianodon hypophthalmus*) infected with *Aeromonas hydrophila*, which produces hemorrhagic septicemia, compared to 13% of survival in the control group [64].

*Vibrio* sp. produce mortality in bivalve larvae and bacteriophages could be used as biocontrol agents in oyster hatcheries. Two different approaches have been described to solve this problem. The first consists on direct phage treatment comprising two phages, which diminished mortality rates from 77.9% in the control group to 28.2% after just 24 h of incubation [65]. However, the second approach focuses on decontaminating microalgae as vectors for *Vibrio* sp. infection of larval cultures. Phage administration in microalgae resulted in significant reduction of *Vibrio* sp. within 2 h, suggesting that feeding larvae with decontaminated microalgae could be a promising preventive method to avoid infection of bivalve larvae [66]. Curiously, in 2019, a study using a heterologous expression vector was performed against *Vibrio parahaemolyticus*. The yeast *Pichia pastoris* X-33 expressed the phage endolysin Vplys60 from bacteriophage qdv001 and the enzyme was shown to inhibit biofilm formation and to reduce mortality rates for the crustacean *Artemia franciscana* [67]. In other studies, a phage treatment with two phages against *Vibrio anguillarum* infection was effective at 72 h in zebrafish larvae [68], and a cocktail of three phages isolated from sewage showed host specificity against eight *Vibrio coralliilyticus* strains and a *Vibrio tubiashii* strain, obtaining a decrease of over 90% in *V. coralliilyticus* compared to the untreated control [69].

These studies reveal that current results are more promising in aquaculture than in farms. More studies are needed to clarify the real sanitary and economic potential of phage-based therapies in the food industry. It is possible that, as it happens in humans, better results could be obtained by mixing phages and antibiotics due to the synergistic effect.

**Table 2 antibiotics-10-00786-t002:** Summary of reviewed studies using phage therapy in animals.

Animal	Infection/Colonization	Bacteria	Phage Therapy	Outcome	References
Chicken	Salmonellosis and colibacillosis	*S. enterica* serovar Kentucky and *Escherichia coli* O119	*Siphoviridae* (10^7^ PFU) against serovar Kentucky and *Podoviridae* (10 PFU) against *Escherichia coli* orally	Reduction of mortality from 30% to 0% in treated group	[41]
Sheep	Rhinosinusitis	*Pseudomonas aeruginosa*	Cocktail of 4 phages (Pa193, Pa204, Pa222, and Pa223) at 10^8^–10^10^ PFU/mL	Reduction of biofilm biomass on sinus mucosa	[42]
Cow	Subclinical mastitis	*Staphylococcus aureus*	Phage K (10^11^ PFU) intramammary infusions for 5 days	3/18 cows were cured compared to 0/20 of control group	[46]
Sheep	Gut	*Escherichia coli* O157:H7	Oral phage KH1 (10^11^ PFU) or DC22 (10^13^ PFU)	No reduction of strain O157:H7	[52,53]
Sheep	Gut	*Escherichia coli* O157:H7	Cocktail of CEV1 (T4-like) and CEV2 (T5-like) orally	Reduction >99% of *Escherichia coli* in the lower intestinal tract	[54]
Sheep	Gut	*Escherichia coli* O157:H7	Cocktail of 8 phages orally	Reduction of fecal *Escherichia coli* O157:H7, but not in the rumen, 24 h after phage administration	[55]
Pig	Nasal colonization	MRSA V0608892/1 strain	P68 (*Podovirus*) and K* 710 (*Myovirus*) in gel	No reduction observed in the nasal mucosa	[56]
Pig	Prevention	*Salmonella enterica*, *Staphylococcus aureus*, *Escherichia coli* and *Clostridium prefringens*	Cocktail of phages orally	Compared to probiotics, phages had better results as growth promoters, improving digestibility, daily weight gain and gain per feed	[57]
Pig	Ex vivo skin infection	*Acinetobacter baumannii*	IME-AB2 (*Myoviridae*) via gel	Reduction of 90% of bacterial counts 4 h post-treatment	[58]
Honeybee larvae	American foulbrood	*Paenibacillus larvae*	Cocktail of phages 1, 5 and 9	Higher survival rates in hives of treated groups including prophylactic benefits	[60,70]
Common carp	Sepsis	*Citrobacter freundii*	IME-JL8 (*Siphoviridae*)	Decreased pro-inflammatory cytokines and protection of fish from infection when phages were administered 1 h after bacteria, but not after 24 h	[61]
Zebrafish	Sepsis	*Edwardsiella tarda*	ETP-1 for 12 d	Increment of survival from 18% to 68%	[62]
North African catfish	Ulcerative lesions	*Pseudomonas aeruginosa*	Single phage for 8–10 d	7-fold reduction of ulcerative lesions	[63]
Vietnamese striped catfish	Hemorrhagic septicemia	*Aeromonas hydrophila*	Φ2 and Φ5	Increment of survival from 13% to 100%	[64]
Bivalve larvae	Infection	*Vibrio* sp.	Cocktail of Φ5, Φ6 and Φ7	Reduction of mortality from 77.9% to 28.2%	[65]
Microalgae food of bivalve larvae	Infection	*Vibrio harveyi*	Cocktail of Φ1, Φ2, Φ3 and Φ4	10 times reduction of bacteria after 2 h	[66]
Zebrafish larvae	Infection	*Vibrio anguillarum*	VA-1 phage	Mortality rate after 72 post-infection was reduced from 17€ to 3%.	[68]
Larval Pacific oysters	Infection	*Vibrio coralliilyticus*	Cocktail of vB_VcorM-GR7B, vB_VcorM-GR11A, and vB_VcorM-GR28A	Mortality reduction of >90% respect to the control group	[69]

Abbreviations: PFU, plaque-forming units; MRSA, methicillin-resistant *Staphylococcus aureus*.

## 4. Phage Therapy for Plants

Different pathogenic bacteria produce significant economic losses in plant production worldwide. This section focuses on recent advances in phage use against pathogens infecting economically relevant plants such as potatoes, tomatoes, cherries, onions, kohlrabies and melons.

*Potatoes*: *Pectobacterium atrosepticum* is a pathogenic bacterium causing soft rot disease and blackleg disease. A cocktail of six phages infected 93% of tested strains and succeeded for biocontrol by decreasing disease incidence (61%) and severity (64%) [71]. Another study treated a mixed infection caused by two different *P. atrosepticum* strains with a cocktail of three bacteriophages and the results showed that the average weight of rotten tissue decreased significantly from 5.39 g in infected plants to 0.31 g in treated tubers [72]. Semi-in planta potato bioassays showed that a cocktail of six phages were able to suppress the growth of a mix of *P. atrosepticum* and *P. carotovorum* subsp*. carotovorum* against soft rot development [73]. Curiously, another study described that phage Pc1 infects *P. carotovorum* subsp. *carotovorum* more efficiently when zinc is not present in the medium, suggesting that inorganic composition of soil is relevant when phage therapy is considered for biocontrol [74]. On the contrary, a Tasmanian potato farm study showed the protective effect of beneficial streptomycetes in soil and pointed that in case of treating the pathogenic strains of *Streptomyces* with phage therapy, a preliminary host range analysis should be performed since a deleterious effect against beneficial streptomycetes might produce opportunistic fungal infections [75]. In a different studio, a cocktail of six phages was used to combat the potato pathogen *Dickeya solani* in soft rots. The cocktail was able to reduce the disease incidence in infected tubers from 93.3% to 48.9% and decrease the diseased tissue by 75.3% [76]. Similar results were found previously with T4-related phages. The treatment of rotting of potato tubers with one phage decreased weight of rot from 4 g to 0.5 g at MOI of 100 [77]. Interestingly, the injection of six phages prior infection protected 80% of potato plants from the *Ralstonia solanacearum* wilt. Phage treatment of contaminated soil also reduced more than 5-fold the presence of this pathogenic bacteria compared to the control soil one week after phage spraying. Efficiency was shown to depend on timing of phage administration, suggesting that phage administration should be performed just after the first sign of bacterial wilt [78].

*Tomatoes*: Several studios with tomato plants are available in the literature. The application of phage PE204 to the root system of tomato plants completely inhibited bacterial wilt caused by *R. solanacearum* [79]. Phages isolated from river water also reduced significantly bacterial wilt and cocktails were the most effective candidates [80]. A greenhouse experiment with combinations of phages against *R. solanacearum* suggested that cocktails of phages select slow-growing resistant bacteria which reduces the severity of the disease [81]. Importantly, a seedling-based method has been recently developed by mixing phages and tomato seedlings in sterile conical tubes before applying *Pseudomonas syringae* to screen phage effectiveness. The authors propose this method before choosing phage candidates in phage biocontrol [82].

*Cherries*: A treatment with thirteen individual phages or two cocktails produced a reduction in the disease progression and a decrease of 15–40% of *P. syringae* in cherry leaves [83].

*Onions*: Recently, a phage-biocontrol study was performed against soft rot caused by *Pectobacterium* sp. in onions using field trials. The results showed significant higher number of plants in the treatments compared to the positive controls, with concomitant increased bulb and foliage mass and also reduced soft rot disease symptoms [84].

*Kohlrabies*: The administration of a single phage at a MOI of 10 was able to reduce black rot disease symptoms due to *Xanthomonas campestris* pv. *campestris* up to 45% [85].

*Melons*: *Acidovorax citrulli* causing fruit blotch was treated with a single phage and 27% of disease severity was shown compared with 80% of disease of the control group, moreover, phage was detected by PCR in foliar tissues 8 h after phage addition to the soil [86].

Summarizing, the use of phage-biocontrol shows a certain effect in vegetables, mainly when cocktails are administered in a short period time after infection. If this strategy results beneficial, it could be administered in the irrigation water to help decrease losses caused by pathogenic bacteria in cultures of economic relevance.

## 5. Phages on Surfaces

Bacteria are able to attach different surfaces as glass, metals, polymers, foods, as well as to other organisms [87]. The greatest risk of food contamination resides on food-contact surfaces. For this reason, biofilms are a big deal in food industry, since they can spoil the equipment and contaminate food, increasing production costs [88]. The interactions between bacteria and food-processing surfaces begins with a non-specific adhesion and ends with specific adhesions and the biofilm formation [89].

*Salmonella* fimbriae facilitate attachment and the presence of cellulose enhances biofilm formation on certain abiotic surfaces [90]. In 2019, Islam et al. isolated three broad-ranged lytic phages, LPSTLL, LPST94 and LPST153, from environmental water samples. The cocktail reduced *Salmonella* biofilms by 44–63% on 96-well microplates. On food-processing surfaces such as stainless steel the cocktail was able to reduce biofilms cells up to 6.42 log CFU [91]. Remarkably, Sadekuzzaman et al. showed that bacteriophages reduced *Salmonella* in biofilms after only two hours of treatment by 3 and 2 log CFU/cm^2^ on stainless steel and rubber, respectively, while adhered viable cells on lettuce were reduced around 1 log CFU per unit [92]. Gong et al. exhibited that a cocktail of six phages was able to diminish by 84.2% the *Salmonella* population on the boots of workers (which is relevant to prevent re-contamination of rendered meals) in a rendering-processing environment. This reduction increased in combination with sodium hypochlorite (92.9%) and scrubbing (93.2%) after a treatment three times for one week [93]. Interestingly, it has been shown a synergistic effect in the combination of bacteriophages and chlorine with a reduction of biofilm growth by 94% and the ability to remove pre-existing biofilms by 88%, whereas chlorine alone could not eliminate them [94].

*Pseudomonas* is the most frequently reported genus of the bacteria found after sanitation on food processing surfaces. This genus is able to resist in niches with nutrients, surface materials, temperatures and stress factors that are problematic for other bacteria, such as machines, floors, drains or stainless steel [95]. Magin et al. tested 14.1 and LUZ7 phages isolated from drinking and thermal water against 24 h old biofilms produced by *P. aeruginosa* PAO1 and D1 strains. Results showed that phage treatment produced a reduction of 1.7 log CFU/cm^2^ of bacteria in biofilms formed on stainless-steel surface compared with untreated biofilms [96].

On the other hand, *E. coli* can attach to a variety of surfaces including stainless steel, teflon, glass, polystyrene, polypropylene, PVC and biotic surfaces, which are commonly employed in food industry. Wang et al. tested the AZO145A phage against the Shiga toxigenic *E. coli* O145:H25 strain, known to be a strong biofilm former, on stainless steel coupons. Bacteriophage addition on biofilms grown during 24, 48 and 72 h was able to reduce cells 2.9, 1.9 and 1.9 log CFU/coupon, respectively, compared to the control [97].

Overall, bacteriophages show great promise in decreasing the formation of new biofilms, but most importantly, in removing pre-existing ones in combination with other agents such as bleach.

## 6. Bacteriophages in Bioremediation

Most of the hydrocarbons contaminating water can be used as a source of carbon by a large number of bacteria such as *P. aeruginosa*, which is capable of degrading monoaromatic hydrocarbons [98] or with species of *Rhodococcus* genus, capable of degrading cyclohexane [99]. Recent studies have identified bacteria from more than 79 genera capable of degrading petroleum hydrocarbons [100]. The employment of microorganisms in bioremediation is based on the microbial loop. The main role of the microbial loop is the fast CO_2_ production and the recycle of nitrogen and phosphorus in the environment [101]. Rosenberg et al. tested the efficiency of two bioreactors with bacteria/phage combinations at different concentrations for the treatment of drainage water from an Israeli oil terminal. Their study showed a total organic carbon reduction of 85% in the bioreactor with less bacteriophages and 90% of reduction with a higher phage concentration compared to the control, which supports the concept of a phage-driven microbial loop [102]. Phages can immobilize some nitrogen or phosphorus, but the main impact is caused by the bacterial lysis and the release of constituents into the water as dissolved organic Carbon, thereby increasing the bacterial growth [102,103].

On the other hand, phages have been also tested to help in the treatment of activated sludge bulking and foaming. Khainar et al. isolated specific bacteriophages against nocardioforms on active sludge process. The activity of three phages applied in a cocktail at the lab scale reactor reduced foam formation [104]. Choi et al. isolated a bacteriophage from sewage infecting *Sphaerotilus natans*, known to cause filamentous bulking in wastewater treatment systems, and their results showed that phage application diminished the sludge volume index and turbidity of the supernatant, indicating that phages can be used in this concern too [105].

## 7. Discussion

Phages are promising candidates in the fight against MDR bacteria. Recent studies report that phage treatment is able to reduce bacterial load and biofilm formation in biotic and abiotic media, indicating that this approach can be useful in biotechnology. However, one of the main concerns when considering this alternative is the narrow host range of most of the phages. This can be reverted almost completely with the use of phage cocktails. Moreover, cocktail use decreases the appearance of phage-resistant strains. High MOI and a rapid administration have been shown to increase successful rates of phage therapy in controlled experiments, however, in the real practice these two parameters cannot be determined.

Two different approaches of phage therapy have been proposed depending on the goal, humans and non-humans. In the first case, phage therapy is administered usually in combination with antibiotics due to their synergistic effect, while in the second case studies are performed typically only with phages to avoid antibiotics. Lessons learnt for human use could be helpful to succeed in non-human practice. Successful case reports in humans usually have administered cocktails of phages combined with antibiotics in multiple doses. Therefore, the combination of cocktails and low levels of antibiotics could improve the results of ineffective phage therapy in non-human use. On the other hand, there are cases in which neither antimicrobials nor phages can solve the injury, such as the case of toxigenic strains in which the harmful effect is due to the toxin. A less explored alternative is the induction of prophages that are latent in the bacterial genomes [106] with compounds like EDTA, sodium citrate [107], glycolic acid, N-acetyl cysteine, vinegar or plant extracts like stevia [108], which would solve issues such as the host range restriction or bacterial resistance, and would improve the reaching to intracellular bacteria [106].

In conclusion, further research is necessary to elaborate standard protocols in each specific field, including farms, aquaculture, surfaces or bioremediation in terms of timing, administration or cocktail composition, although the current phage products available in the market show that this alternative is already a real choice in biocontrol.

## Figures and Tables

**Figure 1 antibiotics-10-00786-f001:**
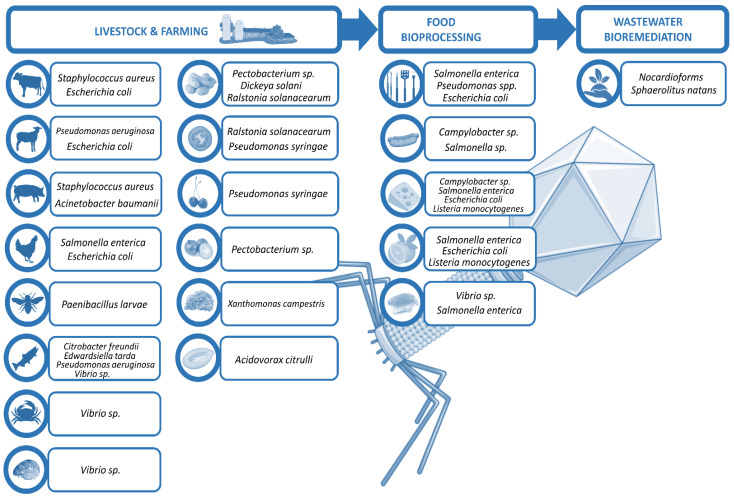
Scheme of phage utilities in biocontrol.

**Table 1 antibiotics-10-00786-t001:** List of approved and commercially available bacteriophage products.

Company	Phage Product	Pathogen
Micreos Food Safety (The Netherlands)	PhageGuard Listex	*Listeria* sp.
	PhageGuard S	*Salmonella enterica*
	PhageGuard E	*Escherichia coli* O157:H7
Intralytix (USA)	ListShield	*Listeria monocytogenes*
	SalmoFresh	*Salmonella enterica*
	ShigaShield	*Shigella* sp.
	EcoShield PX	*Escherichia coli*
Arm & Hammer (USA)	Finalyse SAL	*Salmonella enterica*
	Finalyse	*Escherichia coli* O157:H7
Omnilytics (USA)	BacWash	*Salmonella enterica*, *Escherichia coli* O157:H7
	AgriPhage	*Xanthomonas campestris*, *Pseudomonas syringae*
APS Biocontrol Ltd. (UK)	Biolyse-PB	*Erwinia* sp., *Pectobacterium* sp., *Pseudomonas* sp.
Proteon Pharmaceuticals SA (Poland)	Bafasal	*Salmonella enterica*
	Bafador	*Pseudomonas* sp., *Aeromonas* sp.
FINK TEC GmbH (Germany)	Secure Shield E1	*Escherichia coli*
Brimmedical (Georgia)	PYO Phage	*Staphylococcus* sp., *Escherichia coli*, *Streptococcus* sp., *Pseudomonas* sp., *Proteus* sp.
	Intesti Phage	*Shigella* sp.*, Salmonella enterica*, *Staphylococcus* sp., *Proteus* sp., *Escherichia coli*, *Pseudomonas aeruginosa*
	SES Phage	*Staphylococcus* sp., Enteropathogenic serotypes of *Escherichia coli*, *Streptococcus* sp.
	EnkoPhagum	*Salmonella enterica*, *Shigella* sp., Enteropathogenic serotypes of *Escherichia coli*, *Staphylococcus* sp.
	Fersisi Phage	*Staphylococcus* sp., *Streptococcus* sp.
	Mono-phage	*Staphylococcus* sp., *Escherichia coli*, *Streptococcus* sp., *Enterococcus* sp.*, Pseudomonas aeruginosa*, *Proteus* sp.

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
