# Peer review of "Phages in Food Industry Biocontrol and Bioremediation"

_antibiotics, 2021, doi:10.3390/antibiotics10070786_

Round 1

Reviewer 1 Report

The review article on phages in food industry biocontrol and bioremediation is a well-writted paper. The authors have summarized state-of-the-art in this field, thogh they have discussed only selected papers which were published in this field. This is, however, reasonable as the literature in this topic is very reach, thus, some selection was necessary, indeed. I have only several minor points which should be addressed during revision.

  1. Please, explain abbreviations used in the Abstract (MDR, WHO). In fact, they are explained in the body text, however, Abstract is a separate part of the paper, published without the full text in various data bases, thus, it should be self-explanatory.
  2. The first chapter (Introduction) is very poor in citations. Only 3 papers are cited in the whole chapter which is very insufficient. In fact, in a review paper, every specific information should be supported by relevant reference(s) which is crucial for readers of such a paper. Otherwise, the information is useless without giving the primary source (especially if it is a fact, not authors' opinion). The same problem occurs in chapter 2 (not subsequent subchapters which are fine) - for example, what CDC document the authors refer to (line 59)?
  3. Table 1. Please, be consistent with names of microorganisms. Use either full names or abbreviated forms throughout the table (I prefer full names). If only genus is known (without determination of species), use the genus name followed by "sp.".
  4. Use full names of bacteria in titles of (sub)chapters, e.g. chapter 2.3.
  5. Be careful with names of Salmonella species and serovars. For example, Salmonella kentucky is not a species! Kentucky is a serovar, and should be written accordingly in the name of this microorganism (see microbial nomenclature guide).
  6. Some references in the list of cited literature are incomplete. Ref. 3 - no journal name and no information on the year of publication are provided. Ref 8 - No information at all, apart from the title. What is it? If it is a book, provide title, year of publication and publisher. If web page, provide web address. Ref. 25 - No information at all, apart from the title. What is it? If it is a book, provide title, year of publication and publisher. If web page, provide web address. Ref. 26 - No information at all, apart from the title. What is it? If it is a book, provide title, year of publication and publisher. If software, provide company name. If web page, provide web address. Ref. 31 - No information at all, apart from the title. What is it? If it is a book, provide title, year of publication and publisher. If web page, provide web address. 
  7. The paper would benefit from including more figures/schemes. However, this is optional.

Author Response

Comments and Suggestions for Authors

The review article on phages in food industry biocontrol and bioremediation is a well-written paper. The authors have summarized state-of-the-art in this field, though they have discussed only selected papers which were published in this field. This is, however, reasonable as the literature in this topic is very reach, thus, some selection was necessary, indeed. I have only several minor points which should be addressed during revision.

  1. Please, explain abbreviations used in the Abstract (MDR, WHO). In fact, they are explained in the body text, however, Abstract is a separate part of the paper, published without the full text in various data bases, thus, it should be self-explanatory.

RESPONSE: We thank the suggestion of the reviewer, we have added the explanation of the two abbreviations in the Abstract section (lines 10 and 12).

  1. The first chapter (Introduction) is very poor in citations. Only 3 papers are cited in the whole chapter which is very insufficient. In fact, in a review paper, every specific information should be supported by relevant reference(s) which is crucial for readers of such a paper. Otherwise, the information is useless without giving the primary source (especially if it is a fact, not authors' opinion). The same problem occurs in chapter 2 (not subsequent subchapters which are fine) - for example, what CDC document the authors refer to (line 59)?

RESPONSE: We agree with the reviewer. More citations have been added to the introduction section (lines 34, 38, 42, 46 and 49) and in chapter 2 (line 61).

  1. Table 1. Please, be consistent with names of microorganisms. Use either full names or abbreviated forms throughout the table (I prefer full names). If only genus is known (without determination of species), use the genus name followed by "sp.".

RESPONSE: We agree with the reviewer and we have added full names in Table 1.

  1. Use full names of bacteria in titles of (sub)chapters, e.g. chapter 2.3.

RESPONSE: Short names of bacteria have been replaced by full names in titles.

  1. Be careful with names of Salmonella species and serovars. For example, Salmonella kentucky is not a species! Kentucky is a serovar, and should be written accordingly in the name of this microorganism (see microbial nomenclature guide).

RESPONSE: We agree with the reviewer and we have modified the name to Salmonella enterica serovar Kentucky (lines 197 and 198).

  1. Some references in the list of cited literature are incomplete. Ref. 3 - no journal name and no information on the year of publication are provided. Ref 8 - No information at all, apart from the title. What is it? If it is a book, provide title, year of publication and publisher. If web page, provide web address. Ref. 25 - No information at all, apart from the title. What is it? If it is a book, provide title, year of publication and publisher. If web page, provide web address. Ref. 26 - No information at all, apart from the title. What is it? If it is a book, provide title, year of publication and publisher. If software, provide company name. If web page, provide web address. Ref. 31 - No information at all, apart from the title. What is it? If it is a book, provide title, year of publication and publisher. If web page, provide web address.

RESPONSE: We apologize for the mistake. We have added the missing information of the references mentioned above (current numbers of references: 3, 14, 31, 32, and 37, respectively).

  1. The paper would benefit from including more figures/schemes. However, this is optional.

RESPONSE: As suggested by the reviewer, we have included a new table (Table 2), which summarizes the reviewed research of phages in animals.

Reviewer 2 Report

This review summarizes the application of bacteriophages in bioremediation, but there are few relevant examples.

Author Response

Comments and Suggestions for Authors

This review summarizes the application of bacteriophages in bioremediation, but there are few relevant examples.

RESPONSE: We agree with the reviewer that the review discusses only selected papers since the literature in this topic is very rich. We focused on in vivo experiments and we excluded the in vitro ones. We have clarified this criterium in the Introduction section (line 52).

Reviewer 3 Report

The manuscript “Phages in food industry biocontrol and bioremediation” by Pablo Cristobal-Cueto et al. summarizes the state of phage application for food safety. This is a thorough analysis that includes bacteria which cause problems throughout the food industry pipeline, from farming to processing and wastewater remediation. The text is detailed and the figure and table are excellent, but additional presentation items would be helpful in digesting the amount of information.

Primarily, an extremely useful resource would be an additional table outlining the types of phages used in the Table 1 products or which are discussed in the text (e.g. Felix-O1a and S16 are mentioned on line 87). This could identify trends of which phages are most useful: T4 phages? Felix-O1 phages? P22 phages?

Minor points:

  1. Phage P22 is mentioned on line 74 for biocontrol: was this an obligately lytic mutant?
  2. The images/icons in the Food Bioprocessing section of Figure 1 are a bit difficult to see. The style of the animals is very clear, but the other images are more challenging to interpret.
  3. It would be helpful if the definition of “microbial loop” on lines 393-394 were moved up to line 388.

Author Response

Comments and Suggestions for Authors

The manuscript “Phages in food industry biocontrol and bioremediation” by Pablo Cristobal-Cueto et al. summarizes the state of phage application for food safety. This is a thorough analysis that includes bacteria which cause problems throughout the food industry pipeline, from farming to processing and wastewater remediation. The text is detailed and the figure and table are excellent, but additional presentation items would be helpful in digesting the amount of information.

Primarily, an extremely useful resource would be an additional table outlining the types of phages used in the Table 1 products or which are discussed in the text (e.g. Felix-O1a and S16 are mentioned on line 87). This could identify trends of which phages are most useful: T4 phages? Felix-O1 phages? P22 phages?

RESPONSE: We agree with the reviewer that this information would be beneficial, however, we did not introduce the type of phages in Table 1 because the websites of the commercial products do not mention them. We only have the information published in some papers that are already in the main text. In addition, we have introduced Table 2 in order to summarize relevant information of experiments in animals and we have included the type of phages when it was described.

Minor points:

  1. Phage P22 is mentioned on line 74 for biocontrol: was this an obligately lytic mutant?

RESPONSE: Phongtang et al. used the phage P22 ATCC 97541 without mentioning any genetical manipulation. We have introduced this information in the text (line 77).

  1. The images/icons in the Food Bioprocessing section of Figure 1 are a bit difficult to see. The style of the animals is very clear, but the other images are more challenging to interpret.

RESPONSE: We agree with the reviewer and clearer images have been included in Figure 1.

  1. It would be helpful if the definition of “microbial loop” on lines 393-394 were moved up to line 388.

RESPONSE: As suggested by the reviewer, we have changed the order of the mentioned sentence (current lines 395-396).